# Optimizing ctDNA: An Updated Review of a Promising Clinical Tool for the Management of Uveal Melanoma

**DOI:** 10.3390/cancers16173053

**Published:** 2024-09-01

**Authors:** Mar Varela, Sergi Villatoro, Daniel Lorenzo, Josep Maria Piulats, Josep Maria Caminal

**Affiliations:** 1Department of Pathology, Hospital Universitari de Bellvitge, 08907 L’Hospitalet de Llobregat, Barcelona, Spain; svillatoro@bellvitgehospital.cat; 2Laboratori Core d’Anàlisi Molecular, Hospital Universitari de Bellvitge—Institut Català d’Oncologia, 08907 L’Hospitalet de Llobregat, Barcelona, Spain; 3Ophthalmology Department, Hospital Universitari de Bellvitge, 08907 L’Hospitalet de Llobregat, Barcelona, Spain; dlorenzo@bellvitgehospital.cat; 4Institut d’Investigació Biomèdica de Bellvitge (IDIBELL), 08908 L’Hospitalet de Llobregat, Barcelona, Spain; jmpiulats@iconcologia.net; 5Medical Oncology Department, Institut Català d’Oncologia, 08908 L’Hospitalet de Llobregat, Barcelona, Spain

**Keywords:** liquid biopsy, circulating tumor DNA, uveal melanoma, precision medicine, oncology, molecular techniques

## Abstract

**Simple Summary:**

Liquid biopsy based on the detection of circulating tumor DNA (ctDNA) is a well-consolidated tool to guide treatment decision-making and monitor patients with cancers other than uveal melanoma. The aim of this study was to explore the current technical possibilities of liquid biopsy to detect ctDNA in patients with uveal melanoma, with a particular focus on the clinical setting, to provide an overview of the current use of this technique.

**Abstract:**

Uveal melanoma (UM) is the most common primary malignant intraocular tumor in adults. Distant metastasis is common, affecting around 50% of patients. Prognostic accuracy relies on molecular characterization of tumor tissue. In these patients, however, conventional biopsy can be challenging due to the difficulty of obtaining sufficient tissue for the analysis due to the small tumor size and/or post-brachytherapy shrinkage. An alternative approach is liquid biopsy, a non-invasive technique that allows for real-time monitoring of tumor dynamics. Liquid biopsy plays an increasingly prominent role in precision medicine, providing valuable information on the molecular profile of the tumor and treatment response. Liquid biopsy can facilitate early detection and can be used to monitor progression and recurrence. ctDNA-based tests are particularly promising due to their ease of integration into clinical practice. In this review, we discuss the application of ctDNA in liquid biopsies for UM. More specifically, we explore the emerging technologies in this field and the advantages and disadvantages of using different bodily fluids for liquid biopsy. Finally, we discuss the current barriers to routine clinical use of this technique.

## 1. Introduction

Liquid biopsy is less invasive and easier to perform than surgical biopsy. This technique involves the analysis of blood samples or other bodily fluids (e.g., urine, saliva, breast milk, cerebrospinal fluid, aqueous humor, and tears, among others) to obtain valuable tumor-related information. Liquid biopsy allows for the comprehensive molecular profiling of a wide range of tumors, thus facilitating precision oncology. Liquid biopsy is currently used in several different tumor types, where it has been shown to improve diagnostic accuracy and treatment selection by helping to select targeted therapies [1]. During follow-up, it can help to monitor disease progression. Significant efforts are underway to use liquid biopsy for early cancer detection and to identify minimal residual disease [2].

Liquid biopsy requires the isolation of cancer cells or cancer cell-derived components from bodily fluids for subsequent analysis. The primary analytes include circulating tumor cells (CTCs), circulating tumor DNA (ctDNA), ctRNA, circulating microRNA (miRNA), and extracellular vesicles.

In the past decade, numerous studies have demonstrated the value of liquid biopsy in confirming diagnosis without the need for conventional biopsy. Those studies have also shown the prognostic capacity of liquid biopsy, which can be used for the early detection of metastatic spread and to monitor treatment response, as reviewed in [3]. However, liquid biopsy is mainly used for translational research and clinical trials. The role of liquid biopsy in routine clinical care is still developing.

The current gold standard for detecting mutations is conventional tissue biopsy, but multiple biopsy sampling might pose a risk of extraocular dissemination [4]. In this review, we focus on the potential of liquid biopsy to detect mutations through the analysis of ctDNA in the context of routine clinical practice [5]. One potential advantage of liquid biopsy over conventional biopsy is that the latter is based on tissue obtained from a single location, which may not reflect the tumor’s full heterogeneity and genomic complexity [6]. Liquid biopsy of ctDNA could overcome that limitation. Imperial et al. [7] observed significant discordance between next-generation sequencing (NGS) of tissue biopsies and ctDNA analysis of liquid biopsies in somatic hotspot mutations, a finding that denotes significant tumor heterogeneity in these malignancies.

The success or failure of translating biomarker determination in liquid biopsy from the laboratory to routine clinical practice depends on various factors. These include access to sophisticated technical resources in clinical settings, reasonable response times to ensure appropriate patient management and treatment, the high cost of molecular tests, and the availability of commercial solutions for in vitro diagnostics approved by the U.S. Food and Drug Administration (FDA) and/or In Vitro Diagnostic Regulation (IVDR) marking under the new European Medicines Agency (EMA) regulation. The first liquid biopsy test was approved by the FDA in 2013 (CellSearch^®^ CTC). Since then, numerous other tests—all of which use ctDNA—have been approved for targeted molecular drugs and multigene panel assays of liquid biopsy as companion diagnostics in different tumor types [8,9]. ctDNA-based tests are more common than CTC-based tests because more technologies are available to isolate these analytes. In addition, CTCs are more fragile and harder to transport than plasma. In short, the main advantage of ctDNA over CTCs is that the former is easier to handle in routine clinical practice.

Uveal melanoma (UM) is the most common primary intraocular tumor in adults. UM has a strong capacity to metastasize, and approximately 50% of patients develop distant metastases. The most common metastatic site is the liver, with 90% of patients with metastatic disease presenting liver involvement [10]. In this context, the aim of the present study was to review and summarize current knowledge on the clinical application of ctDNA-based liquid biopsy in patients with UM. This review primarily focuses on recent technological advances, the most relevant clinical applications, and limitations related to using ctDNA analysis in routine clinical practice.

## 2. Sources of Liquid Biopsies

While blood is a key source of analytes, other fluids in the human eye can be used for liquid biopsy, including the vitreous humor, aqueous humor (anterior chamber), and tears. Although the process of obtaining vitreous and aqueous humor samples is not as minimally invasive as with blood, these samples are being successfully used to characterize UM tumors [11].

### 2.1. Blood-Based Liquid Biopsy in Uveal Melanoma

In UM, tumor cell dissemination is hematogenous, which means that samples are easy to obtain, making blood an ideal source of analytical material. However, using plasma as a source of analytes can be challenging because it is difficult to detect ctDNA in patients with primary UM. Detection rates are highly variable and influenced by factors such as tumor size, the technology used, and the timing of sample collection [12]. In this regard, Kim et al. [13] demonstrated that brachytherapy increases the presence of ctDNA in the plasma of patients with UM, a finding that suggests that the timing of sample collection during treatment could play a role in determining the success or failure of ctDNA detection. A recent study by de Bruyn et al. showed that ctDNA levels in blood were elevated in metastatic UM and thus easier to detect at this stage than in localized disease [14].

### 2.2. Vitreous Liquid Biopsy in Uveal Melanoma

Most published studies that have evaluated the molecular characterization of UM using vitreous humor samples have focused on proteomic profiling. Vitreous fluids can be safely collected at the beginning of a vitrectomy [15]. Retinal proteins filter into the vitreous, and the proteomic analysis of biomarkers concentrated in the liquid compartments adjacent to the tumor provides valuable prognostic information. This technique could be used to identify new biomarkers or therapeutic targets to potentially improve the quality of life for patients with UM [16].

The vitreous humor has proven to be an adequate source of ctDNA to monitor and treat other ocular pathologies, such as vitreoretinal lymphoma, where massive sequencing techniques have been applied to obtain a more comprehensive molecular profile [17,18]. However, this approach has not been widely used in UM to date. Although some preliminary studies have reported positive results from evaluating ctDNA in vitreous fluid, more research is needed [19]. More specifically, we need to obtain a better understanding of whether this type of sample can provide key molecular information about the primary tumor. We also need to determine if it can be used to estimate the risk of metastatic dissemination and to monitor the course of the disease without the need for direct tumor biopsy [20].

### 2.3. Aqueous Humor Liquid Biopsy in Uveal Melanoma

UM can involve the choroid, iris, and ciliary body. The iris and ciliary body are located close to the aqueous humor, which makes this a good candidate for sampling. The aqueous humor has numerous advantages as a source of ctDNA for liquid biopsy: (1) it is more accessible than the vitreous humor; (2) the sample can be obtained with less specialized instrumentation; (3) the sample can be collected in an outpatient setting; and (4) repeat sampling can be performed, thus enabling individualized surveillance following treatment.

Pike et al. [21] conducted a multicenter study to explore the potential clinical correlation between analytes and tumor features. Those authors used fluorometry to quantify unprocessed analytes (dsDNA, miRNA and proteins) from aqueous humor samples at diagnosis and after brachytherapy, finding significantly higher concentrations of dsDNA in samples obtained from UM patients than in controls. Although the analytes were quantifiable even in eyes with smaller, less advanced tumors, samples obtained from eyes with more advanced stages and larger tumors had higher concentrations of analytes.

Recently, Im et al. [22] demonstrated that the aqueous humor of patients with UM contains sufficient ctDNA to perform complete genetic tests. The yield was significantly higher in samples obtained after brachytherapy and anterior tumors (e.g., ciliary body). Proteomic studies have shown that analysis of aqueous humor samples can help establish the risk of metastasis in patients with UM [23,24]. These findings are important as this technique could be used in future studies to analyze ctDNA.

## 3. The Molecular Landscape of Uveal Melanoma

Given the presence of molecular alterations in UM, molecular techniques can be applied to detect ctDNA. The molecular characterization of tumor tissue in UM is essential to ensure an accurate prognosis, which is influenced by the presence or absence of specific driver mutations and somatic copy number alterations (SCNA). Unfortunately, the quantity of primary tumor tissue available for molecular characterization in UM is limited, especially in small UM and after eye-sparing irradiation. In this context, liquid biopsy may be an excellent alternative to avoid the risks and complications of intraocular biopsy. Liquid biopsy can help to identify the genomic mutations involved in the primary tumor or metastatic lesions, monitor response to specific treatments, and determine whether patients meet the criteria for inclusion in a clinical study [25,26].

Most UM tumors share a very limited repertoire of activating driver mutations in the mitogen-activated protein kinase (*MAPK*) signaling pathway (Gαq). In more than 90% of tumors, the mutations affect the *GNAQ* or *GNA11* genes, which encode the α subunit of G protein-coupled receptors [27]. Alterations are sometimes observed in the *CYSLTR2* (4%) or *PLCB4* (2.5%) genes, but these are much less common [28].

In the *GNAQ* and *GNA11* genes, the most commonly mutated codon is Q209. Less frequently, there may be mutations in the R183 codon or, in rare cases, in the G48 codon. For tumors with *CYSLTR2* mutations, the codon involved is L129. For tumors with mutations in *PLCB4*, the affected codon is D630.

Circulating tumor DNA is shed into the bloodstream when cells undergo apoptosis and necrosis [29], so ctDNA levels rise during and after radiotherapy. In fact, this finding led researchers to explore the viability of quantifying driver point mutations and SCNAs. In this regard, the determination of ctDNA levels in the blood could be used as a biomarker, particularly when the primary tumor is large and metastatic disease is present due to the higher tumor burden [14]. Nonetheless, tumor size is not necessarily a major impediment, as shown by Bustamante et al., who successfully detected ctDNA in 100% of patients with primary UM [30].

UM tumors commonly present secondary mutations that have variable effects on metastatic dissemination. These secondary mutilations mainly occur in the *BAP1* (45%), *SF3B1* (25%), and *EIFAX1* (15%) (BSE) genes in a mutually exclusive pattern [31]. While the presence of *EIF1AX* mutations rarely leads to distant metastases, *SF3B1* mutations in the primary tumor are associated with the development of metastatic disease within 10 years of the primary diagnosis (median, 8.2 years) [32,33]. UM tumors with *BAP1* mutations exhibit a more aggressive phenotype with a higher risk of metastasis in the short term [34].

The mutations present in the *EIF1AX* gene are concentrated in exons 1 and 2 (codons 1–20), mainly missense mutations or deletions. Tumors with mutations in the *SF3B1* gene have a mutational hotspot in the R625 codon; other missense mutations (K666 and K700 codons) have been described, but these appear to be much less common [33].

*BAP1* is a tumor suppressor gene located on chromosome 3p. In UM, biallelic inactivation of *BAP1* occurs due to the mutation of a single allele and the loss of the chromosome containing the other allele. Diverse mutations are seen in *BAP1*, including missense mutations, splice site mutations, in-frame deletions, large deletions, and mutations that cause premature truncation. The absence of a clearly defined hotspot means that sequencing techniques are required to detect a wide range of mutations [35].

The SCNAs in primary UM can help to assess metastatic risk. Several different SCNAs—monosomy 3 (M3) [36], amplification of 8q (four or more copies), and deletion of 1p, 8p, and 16q—have all been associated with an unfavorable prognosis in UM [28,37]. Most UM tumors (83%) with M3 also harbor *BAP1* mutations. In 7% of cases, the cytogenetic profile is characterized by the presence of M3, amplification of 8q, and deletion of 1p or 16q; the presence of these mutations defines a group considered to have an ultra-high risk of developing metastasis within four years [37]. Tumors that are diploid for chromosome 3 and/or exhibit 6p gain are associated with low metastatic risk and tend to harbor *EIF1AX* or *SF3B1* mutations [28] (Figure 1).

## 4. Isolation Techniques for ctDNA

Cell-free DNA (cfDNA) consists of DNA fragments released into the bloodstream through cell death processes (e.g., apoptosis, necrosis, pyroptosis, or autophagy). However, cfDNA can also be actively released by living cells through processes related to exosomes or autophagy.

In cancer patients, a proportion of these cfDNA molecules also come from the tumor (primary or metastasis). The ctDNA to total cfDNA tumor fraction is small, ranging from 0.01% to more than 10%, depending on the type of cancer, tumor load, and tumor metabolism [38]. These ctDNA fragments are usually smaller than the cfDNA released by healthy cells, typically measuring between 90–150 base pairs [39].

Several pre-analytical factors should be considered to ensure optimal performance on liquid biopsy samples, including the type of tube used for sample collection, centrifugation protocol, effect of long-term storage, and multiple freeze and thaw cycles [40,41,42].

Several studies have compared commercial solid-phase extraction methods, including either silica matrix or paramagnetic bead technology, both automated and manual [43,44,45,46]. This solid-phase extraction technique has been successfully used to isolate genomic DNA, but it has efficiency limitations in obtaining low molecular weight DNA [47].

Wang et al. [48] compared the performance of four commercial cfDNA isolation kits for plasma samples. Of those four kits, the QIAamp Circulating Nucleic Acid (QCNA) kit had the best performance when comparing recovery rates of 173 base pair DNA fragments (the maximum length of ctDNA), with a recovery rate of 55.67%.

A round-robin trial involving 14 laboratories was performed to compare the performance of cfDNA extraction methods used in diagnostic laboratories in the Netherlands. That study found that silica membrane-based methodologies had a higher total cfDNA yield, leading the authors to conclude that this technique should be used to detect mutations at low allelic frequency. That study also underscored the significant heterogeneity in pre-analytical workflows at the participating laboratories, which can influence ctDNA detection in clinical practice. Those authors concluded that greater harmonization of extraction workflows is needed to ensure that quantification and detection are sufficiently accurate and sensitive, respectively, to reduce inter-laboratory discrepancies [49].

To overcome the performance limitations of solid-phase extraction methods, a new liquid-phase extraction method has been developed to isolate and purify cfDNA based on aqueous two-phase system (ATPS) isolation [50]. Janku et al. [51] developed a new DNA isolation kit, the PHASiFY MAX cfDNA Extraction Kit (CE, NMPA-certified). The validation study for that kit in plasma samples showed a 60% increase in DNA recovery compared to QCNA. Selection of the most appropriate method for a particular laboratory must take into account numerous variables, including cost, feasibility of automation, processing time, sample type, and the laboratory’s workload capacity.

Many questions related to ctDNA extraction are still unresolved, including the aforementioned divergence in pre-analytical workflows and the lack of standardization for analytical applications. Moreover, the wide variety of commercial products for sample and ctDNA collection makes it difficult to compare laboratories and determine reproducibility [52]. The studies carried out to date to compare and promote process standardization have mainly focused on plasma samples. Comparisons of other types of liquid biopsy samples are scant.

## 5. Molecular Techniques for ctDNA Detection

A wide range of molecular techniques is available to analyze ctDNA. In this review, we discuss those techniques that use polymerase chain reaction (PCR)-based and NGS-based methods (Table 1).

### 5.1. PCR-Based Detection Techniques

PCR-based techniques can detect previously known mutations with a high sensitivity. However, PCR-based techniques are not capable of identifying novel alterations.

#### 5.1.1. Droplet Digital PCR (ddPCR) and Digital PCR (dPCR)

Digital PCR systems assess many single-reaction PCR partitions to evaluate the whole target for a sample result. Several methods can be used to partition samples, including microwell plates, capillaries, oil emulsion, and arrays of miniaturized chambers with nucleic acid binding surfaces. This method uses a fluid in oil where nucleic acids are encapsulated inside droplets; by contrast, in dPCR, the partitioning is performed with a special nanoplate with physical wells on a solid support. End-point PCR is carried out, and samples are processed by flow cytometry, where partitions are fluorescently read one by one as they pass in front of a laser excitation source [69]. ddPCR can detect an average fractional abundance of cfDNA as low as 0.01% [53]. Hashimoto et al. [54] increased the mutation fraction detection to 0.003% using locked nucleic acid-clamp ddPCR.

Both techniques can be used to calculate SCNAs and to identify small insertions and deletions (indels), point mutations and aberrational methylation patterns [70,71], as well as track them down in UM [20,30,72]. However, the detection rate for variants in cfDNA is slightly better with dPCR (86.4–100%) than with ddPCR (58.8–72.7%) [55]. Moreover, droplet-based microfluidics can easily separate biomarkers down to the single-cell, single-molecule, or single-exosome level, with high throughput [53]. In dPCR, the partition type prevents the emulsion and droplets from breaking, reduces run times and risk of contamination, and eliminates the variability in size and coalescence associated with droplets.

Both of these techniques have limitations, mainly because some of the reagents used in the pre-analytical process could interfere with their performance. Another limitation is that they can only detect known mutations and cannot identify new ones.

#### 5.1.2. COLD-PCR

Co-amplification denaturation at low temperature-based PCR (COLD-PCR) is a single-step amplification method that enhances both known and unknown minority alleles during PCR [73]. This technique can be used to analyze DNA isolated from liquid biopsies and can detect mutations from 10–25 ng of DNA with a frequency as low as 0.01%. The detection limit depends directly on the number of molecules in the PCR [56,57].

#### 5.1.3. BEAMing

BEAMing combines PCR with flow cytometry. This technique can detect alterations at levels as low as 0.01%, with a sensitivity of 85% and specificity of close to 100% [58,59]. Although this technique is highly sensitive due to the cost and complex workflow, it is not suitable for routine use in the healthcare laboratory setting, as reviewed in [74].

#### 5.1.4. ARMS-PCR

The amplification refractory mutation system (ARMS) uses sequence-specific PCR primers to amplify target DNA sequences contained in the sample to be interrogated and to detect any mutation involving single base changes or small deletions. In ARMS, amplification is only successful when the target allele is contained in the sample. Thus, the absence of the PCR product determines the diagnosis. This technique can detect allele mutant frequencies as low as 0.015% [60]. Unfortunately, ARMS only can identify previously known mutations [75].

#### 5.1.5. CRISPR/Cas12a Technology

CRISPR/Cas12a-based platform relies on the design of highly specific crRNAs to detect disease-related point mutations. Escalona-Noguero et al. created a fluorescent sensor approach based on this technology in combination with an Allele-Specific PCR (AS-PCR) to detect the UM-associated GNAQ Q209P mutation in ctDNA. This technique detects allele mutant frequencies as low as 3%. However, the plasma from only four patients was tested, and further investigation with a larger cohort is needed [61].

### 5.2. NGS-Based Detection Techniques

NGS ctDNA analysis could provide a more realistic tumor profile. This technique can identify known and unknown mutations, genomic rearrangements, gene fusions, SCNAs, microsatellite instability, and tumor mutation burden. These data are highly useful in helping guide treatment decision-making.

#### 5.2.1. Whole-Genome Sequencing (WGS)

WGS is the process of determining the complete DNA sequence of the genome, including both the coding and non-coding regions. However, identifying point mutations using WGS of liquid biopsy samples is challenging, especially because the conventional mutation calling software used in this process cannot identify small alterations due to low sequencing coverage and low mutant allelic fractions [76]. For this reason, it is advisable to apply new bioinformatics approaches, such as machine learning mutation, to detect tumor-related point mutations. Despite the technical difficulty of WGS, Im et al. [22] successfully profile SCNAs in 37 aqueous humor samples using shallow WGS. Christodoulou et al. [77] suggested that WGS (to calculate SNCAs) could be combined with targeted sequencing (to identify point mutations, indels, and fusions) to obtain a more comprehensive picture of the tumor from ctDNA. Unfortunately, the WGS of ctDNA is highly complex and expensive, which makes it difficult to implement in routine clinical practice.

Because of WGS complexity, it has so far only been used to detect variants in primary or metastatic UM tumor tissue [78,79].

#### 5.2.2. WES

Whole exome sequencing (WES) assesses all protein-coding exons of the human genome. The value of WES to identify mutations in tumor tissues has been well-demonstrated. Koeppel et al. [80] compared the sensitivity of WES to targeted sequencing, finding a sensitivity of 92% for WES.

The plasma ctDNA to total cfDNA fraction is highly variable, ranging from 0.01% to 90%. However, concordance between tissue tumor-specific variants and ctDNA variants can be achieved with higher ctDNA fractions. For this reason, ultra-low pass (ULP)-WGS is recommended to evaluate the ctDNA fraction cutoff to ensure reliable WES results [81].

WES of ctDNA has been shown to be both feasible and useful in clinical trials to detect novel genomic alterations that may be associated with tumor progression or treatment resistance [82]. However, this technique may not be applicable in all patients due to high variability in the ctDNA fraction. In these cases, targeted sequencing panels of a small set of actionable genes may be more appropriate for use in routine clinical practice. 

#### 5.2.3. Targeted Deep Sequencing

Targeted deep sequencing can be used to sequence-specific genomic regions simultaneously. Targeted deep sequencing of ctDNA has a somatic variant allele frequency threshold as low as 1%, with a median variant allele fraction of 3.65% [83]. Nevertheless, personalized panels, designed with cutting-edge technologies such as anchored multiplex PCR (AMP™) chemistry, enable strand-specific assessment of loci and accurate detection and quantification of high-confidence cell variants as low as 0.008% [64]. Alsina et al. [84] evaluated targeted NGS in formalin-fixed paraffin-embedded and fine-needle aspiration biopsy specimens from patients with UM, which showed a higher frequency detection threshold (up to 5%).

Targeted deep sequencing is definitely the most widely used NGS technique in routine clinical practice to uncover alterations in UM from ctDNA [85,86].

#### 5.2.4. MAPS

Molecular amplification pools (MAPS) is a targeted sequencing technique based on the amplification and sequencing of a gene panel or selected genomic regions. MAPS was developed to reduce NGS sequencing errors. After amplification, each molecular amplification array is indexed and sequenced. Then, the data on variants are compared to remove recurrent errors. MAPS is able to detect insertions, deletions, and single nucleotide variants (SNVs) in allelic frequencies as low as 0.1%. One study used MAPS to analyze cfDNA in lung cancer patients, finding that both sensitivity and sensitivity were excellent (98.5% and 98.9%, respectively) and comparable to ddPCR [65].

#### 5.2.5. CAPP-Seq

Cancer personalized profiling by deep sequencing (CAPP-Seq) is an economical, ultrasensitive method of quantifying ctDNA. This technique combines the identification of known alterations in ctDNA/cfDNA using large genomic optimized library preparation methods with a bioinformatics approach.

CAPP-seq can assess well-characterized tumor alterations with DNA-specific oligonucleotides to determine patient-specific alterations, tumor heterogeneity, and tumor burden. It can identify indels, SNVs, SCNAs, and other rearrangements, but not fusions [66].

#### 5.2.6. Tam-Seq

Tagged-amplicon deep sequencing (Tam-Seq) is a specific analysis designed to increase the sensitivity of NGS (≈97%) and to detect DNA levels as low as 2% by using primers to tag and identify specific genomic sequences. However, the interrogated sequence must be previously known. Despite this limitation, this technique can be performed quickly and inexpensively because it allows the sequencing of large amounts of DNA molecules [87].

A related technique—evolved TadA-assisted N 6-methyladnosine sequencing (eTAm-Seq)—has been reported to identify variant allelic frequencies > 0.1% by combining pre-amplification with chain separation and amplification in an array. It can also detect specific SNVs, indels, SCNAs, and fusions [67,68].

#### 5.2.7. WGBS-Seq

Whole-genome bisulfite sequencing (WGBS-Seq) enables the comprehensive analysis of DNA methylation patterns, thus providing a single cytosine measurement to detect methylated domains in cancer cells with high accuracy. WGBS requires only a minimal amount of plasma to identify ctDNA methylomes with high specificity and sensitivity [88,89].

## 6. Clinical Use of ctDNA in UM

As a biomarker of UM, ctDNA could play an important role in follow-up. Studies show that ctDNA levels increase as the disease progresses (i.e., higher levels of ctDNA are observed in larger tumors and in metastatic disease). Several studies have correlated ctDNA levels with disease progression, treatment response, and regression [30,72,90,91]. Although ctDNA is not currently used in routine practice, it is commonly applied in basic research and clinical trials (metastatic UM). However, this technique needs to be further refined to improve the sensitivity and specificity (Table 2).

UM is an attractive target for liquid biopsy-based biomarker detection with sensitive techniques such as digital PCR for two main reasons. First, nearly all UM tumors present mutations in the Gαq signaling pathway in a mutually exclusive pattern, limited to a few codons (hotspots). Second, these variants are missense variants. Although the presence of these mutations does not affect prognosis, the use of liquid biopsy to assess and quantify these mutations provides valuable information on the tumor burden [86] and can facilitate early detection of metastatic disease. Moreover, these data allow us to monitor response to pharmacological treatments in clinical trials.

Approximately 50% of patients with UM will develop metastatic disease. To prolong survival, close follow-up consisting of frequent imaging and liver function tests (every 3 to 12 months) is recommended to ensure early detection and treatment. Analysis of ctDNA in plasma is being explored as a potentially useful tool to monitor disease progression and predict metastasis.

Beasley et al. [72] assessed ctDNA as a biomarker in a prospective cohort of patients with UM following treatment of the primary tumor. The main aim of that study was to assess the value of ctDNA to detect early signs of metastasis. In that study, ctDNA was detected in 17 of the 25 patients (68%) who had a radiological diagnosis of metastasis. Moreover, ctDNA was the strongest predictor of overall survival in the multivariate analysis.

Carvajal et al. [92] observed a significant linear relationship between the decrease in serum ctDNA and overall survival. Baseline ctDNA levels were correlated with tumor burden. During treatment, ctDNA levels decreased in more than two-thirds of patients, and larger reductions were associated with longer survival, an association that persisted even in patients with radiological progression.

In a prospective cohort of patients with UM, Le Guin et al. [86] sought to determine whether ctDNA was a suitable biomarker for early detection of metastatic disease. In that study, ctDNA was detected before clinical diagnosis of metastasis in approximately half of the patients, with a lead time ranging from 2 to 10 months.

Phase 1 or 2 clinical trials for UM can use ctDNA for real-time monitoring of treatment response, thus providing rapid data on response or disease progression to facilitate timely patient management [85,93]. Gerard et al. [94] hypothesized that a decrease in ctDNA levels may be a better indicator of response to immunotherapy than radiological response alone. This was especially evident in cases of small metastases because tebentafusp-mediated immune infiltration response may mask a tumor reduction, but a significant decrease in ctDNA may be detectable. In fact, lower ctDNA levels and a greater reduction in these levels during follow-up have been suggested as an early indicator of clinical benefit and longer survival [95].

Determination of chromosomal alteration profiles and mutations in BSE genes in blood biopsy samples can provide important prognostic information about the patient’s clinical course. The detection of SCNAs and gene mutations outside of hotspot regions requires the use of sequencing techniques. NGS techniques, which provide complete molecular results of cytogenetic alterations and mutational profiles, have been successfully applied to other ocular tumors, such as retinoblastoma [96,97].

de Bruyn et al. [14] showed that surface WGS (sWGS) combined with in silico size selection can detect loss of chromosome 3 in the peripheral blood of patients harboring metastatic UM. However, in localized disease, sWGS could not determine the status of chromosome 3 from the blood samples. Sato et al. [98] conducted a pilot study to test the applicability of ULP-WGS of cfDNA in UM, which confirmed the effectiveness of that technique for metastatic tumors, especially for the detection of 8q amplification.

Given the low performance of blood-based liquid biopsy to detect chromosomal alterations in non-metastatic patients, other sources are currently being explored to obtain more informative results. In this regard, aqueous humor samples appear to be more useful in anterior tumors (such as ciliary body melanomas) than in choroidal lesions, especially after radiotherapy. Im and colleagues obtained aqueous humor samples following the completion of radiotherapy in patients with primary ciliary body melanoma. In that study, the SCNAs were detected in 75% of samples (four cases had mutations in the *BAP1* and *GNAQ* genes) [22]. Other authors have evaluated vitreous fluid from UM patients. Nell et al. [20] successfully identified mutations in the wild-type Gαq genes in 60% of samples and also identified genetic alterations in the primary tumor with prognostic significance in 42% of patients. Drop-off digital PCR was used to detect BSE mutations. Two different methodologies (classic and SNP-based) were used to determine SCNAs (chromosome 3p and 8q). The classic approach calculates copy numbers by comparing the target region to a genomically stable reference area located on another chromosome. The SNP-based approach evaluates allelic imbalance between two variants of a heterozygous SNP on the chromosome of interest.

In short, this type of liquid biopsy has many potential clinical applications, which could potentially improve early diagnosis and treatment (Table 3). These techniques could be of value in many ways, including (1) rapid detection of UM in indeterminate small melanocytic choroidal lesions, (2) prognosis and monitoring of the disease, (3) identification of high metastatic risk, (4) determination of treatment response, and (5) for patient stratification for targeted therapy [99,100].

## 7. Future Directions and Conclusions

Liquid biopsy has become a key tool in certain tumor types to select the optimal therapeutic approach. It is a useful diagnostic tool and is also valuable to monitor disease progression. Liquid biopsy is increasingly used for the molecular characterization of tumors and to facilitate patient management in the context of precision medicine. Tumors release many different primary analytes into the bloodstream. However, ctDNA-based tests are the easiest to implement in the clinical laboratory, permitting the use of the most advanced molecular technology, including ddPCR, BEAMing, MAPs, TAm-Seq, CAPP-Seq, WES, WGS, and WGBS-Seq.

The application of WES and multi-gene-targeted panels by NGS allows us to obtain a comprehensive genomic profile of the tumor to simultaneously evaluate multiple mutations. However, this approach could be limited by quality assurance, ethical issues, time, and cost. In addition, the available data suggest that this technique may be less sensitive than other methods, especially compared to dPCR. For instance, Iwama et al. [101] found that dPCR was superior to NGS in detecting EGFR-activating mutations in cfDNA at baseline and at disease progression (81.3% and 100% vs. 71.9% and 60.0%, respectively). A meta-analysis performed to assess the relative accuracy of different techniques to detect *KRAS* mutations in cfDNA samples from colorectal cancer patients found that dPCR was more sensitive than NGS (81% vs. 65% respectively), suggesting that the digital platform is more accurate as reviewed in [102].

Due to the molecular characteristics of UM, liquid biopsy offers both advantages and disadvantages over other techniques. One important advantage is related to the hematogenous dissemination of tumor cells, which supports the value of serum-based techniques such as liquid biopsy. In addition, most UM tumors carry point mutations in a few genes, which facilitates their detection with highly sensitive techniques such as ddPCR. Liquid biopsy can also be used to evaluate treatment efficacy in clinical trials and for the early identification of metastatic disease.

Despite the aforementioned advantages, the use of liquid biopsy in routine clinical practice can be challenging. First, from a technical standpoint, the low quantities of ctDNA released from the primary tumor into the bloodstream make prognostic molecular characterization via BSE mutations and chromosomal alterations difficult. However, alternative approaches are currently being studied in an effort to increase the yield of liquid biopsy samples in order to improve the detection of low-frequency molecular alterations. These alternatives include using non-blood sources, such as aqueous or vitreous humor, or sample collection during radiotherapy. The emergence of more sensitive analytical techniques, together with advances in the extraction and purification of ctDNA using liquid-phase methods, may allow us to use liquid biopsy to evaluate indeterminate small melanocytic choroidal lesions in primary tumors and/or in lesions with a low tumor burden.

The optimal methodology has not been defined due, in part, to the high heterogeneity among laboratories in the pre-analytical, extraction, quantification, and analytical techniques. This lack of standardization is important because it can directly affect the detection of molecular alterations, leading to inter-laboratory discrepancies. As a consequence, it is difficult to compare laboratory findings and verify reproducibility. Nonetheless, the International Liquid Biopsy Standardization Alliance (ILSA), which is comprised of organizations and foundations committed to expanding liquid biopsy globally, has been making progress toward standardizing these techniques. They are working to harmonize these techniques to achieve the full “clinical utility” of liquid biopsy in a wide range of contexts [103].

Notwithstanding these efforts towards standardization, it is clear that much work remains to be done. In the context of UM, most of the currently available liquid biopsy techniques are still in the research phase, and more studies are needed to verify the clinical efficacy and safety of these methods. Additionally, given that other ocular fluids besides plasma can be used for liquid biopsy, more research is needed to explore the full potential of these fluids.

In summary, liquid biopsy is a highly promising technique that offers unprecedented possibilities. In the future, liquid biopsy may allow us to identify novel biomarkers of treatment response and improve prognostic accuracy without the need for invasive biopsies. It may also permit detection of metastatic relapse in the early stages. Additionally, ctDNA collection protocols and analytical methods will probably be standardized in the very near future. Further technological advances in this field are expected, which will likely facilitate the discovery of new biomarkers.

In the near future, as analytical protocols become standardized and additional technological advances and discoveries are made, liquid biopsy is expected to become a valuable clinical tool. However, before this tool can be incorporated into routine clinical practice, more improvements are needed. In particular, the sensitivity must be increased in critical contexts to improve detection rates in early-stage disease and cases with minimal residual disease [104]. Prospective clinical trials are needed to explore the true role of cfDNA testing for patient management in the clinical setting.

## Figures and Tables

**Figure 1 cancers-16-03053-f001:**
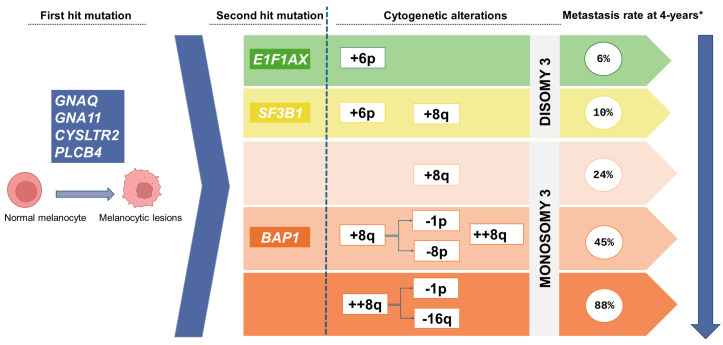
Molecular aspects of UM. After the *GNAQ*, *GNA11*, *CYSLTR2*, or *PLCB4* activating mutation, a second driver event is necessary for malignant transformation in the *BAP1*, *SF3B1*, or *EIF1AX* genes. Chromosomal alterations predict progression towards metastasis. (*****) Data from the study by Lalonde et al., who reported metastasis rates within 4 years at diagnosis by patient subgroups [37].

**Table 1 cancers-16-03053-t001:** Technical characteristics of PCR-based and NGS-based methods to analyze ctDNA.

	Technique	Sensitivity	Specificity	Detection Limit, % ctDNA	Reference
PCR-based	ddPCR	73.3–100%	93.3%	0.10–0.01	Shi et al. [53]
	ddPCR + LNA-Clamp ddPCR	NA	NA	0.003	Hashimoto et al. [54]
	IC3D ddPCR	NA	NA	0.005–0.001	Shi et al. [53]
	dPCR	100%	100%	0.01	Crucitta et al. [55]
	COLD-PCR/E-ice-COLD-PCR	96–97%	96%	0.025–0.01	How-Kit et al., Freidin et al. [56,57]
	BEAMing	85%	99.99%	0.01	Lanman et al., Khagi et al. [58,59]
	ARMS-PCR	77.27%	97.22%	0.015	Khagi et al., Zhang et al. [59,60]
	CRISPR/Cas12a	NA	NA	3	Escalona-Noguero et al. [61]
NGS-based	WGS	97.20%	100%	3	Rickles-Young et al. [62]
	WES	99.8%	99.9%	0.1	Bos et al. [63]
	Targeted NGS	95.7%	99.9%	0.008	Zhao et al. [64]
	MAPS	98.5%	98.9%	0.1	Garcia et al. [65]
	CAPP-Seq	50–100%	96%	0.02	Newman et al. [66]
	Tam-Seq/eTam-Seq	97–99.48%	97–99.99%	2–0.02	Plagnol et al., Gale et al. [67,68]

ddPCR—digital droplet PCR; dPCR—digital PCR; LNA—locked nucleic acid; IC3D ddPCR—droplet digital detection (IC3D) digital PCR system; COLD-PCR—co-amplification denaturation at low temperature-based PCR; E-ice-COLD-PCR—enhanced-improved and complete enrichment-COLD-PCR; BEAMing—beads, emulsion, amplification, and magnetics; ARMS-PCR—amplification refractory mutation system (ARMS) PCR; WGS—whole-genome sequencing; WES—whole exome sequencing; NGS—next-generation sequencing; MAPS—molecular amplification pools; CAPP-NGS—cancer personalized profiling by deep sequencing; Tam-Seq—tagged-amplicon deep sequencing (Tam-Seq); eTam-Seq—evolved TadA-assisted N 6-methyladnosine sequencing.

**Table 2 cancers-16-03053-t002:** Potential advantages and current limitations of clinical use of ctDNA in uveal melanoma.

Advantages	Limitations
Non-invasive genetic uveal melanoma profile;	Demanding technology;
Early detection of malignancy in indeterminate small choroidal melanocytic lesions;	The method has not been fully standardized, and reproducibility is challenging;
High sensitivity to detect early metastatic disease;	Difficult to detect ctDNA in early, low-tumor burden disease;
Useful for prognosis assessment and therapeutic response.	Not clinically validated (although validation studies are underway).

**Table 3 cancers-16-03053-t003:** Summary of recent studies with clinical applications that have used ctDNA analysis in uveal melanoma.

Authors	Fluid Type	Study Population	Number of Patients	System Detection	Detection Rate	Main Findings
Pike et al. [21]	AH	Primary	*n* = 66; samples collected pre- and post-brachytherapy.	Quantification of nucleic acids	Analytes were quantifiable in >70% of diagnostic samples with tumors < 2 mm tall.	AH is a rich repository of analytes. Tumors with poorer prognostic features have increased concentrations of analytes compared with tumors with lower risk.
Im et al. [22]	AH	Primary	*n* = 20; samples taken before or after radiation.	WGS and targeted NGS	SCNAs were found in 75% (6/8) of post-radiation CB samples.	UM, SCNAs and mutations can be identified from the AH, especially in CB tumors.
Bustamante et al. [30]	PB	Primary	*n* = 14.	ddPCR	100% efficiency of UM mutant ctDNA detection.	Potential of ctDNA as a biomarker of early diagnosis and disease progression.
Beasley et al. [72]	PB	Primary and metastatic	Three cohorts: a retrospective cohort of 30 primary tumor patients; a prospective cohort of 37 primary tumors in patients with known mutations, and six patients with metastatic UM.	ddPCR	In a retrospective cohort, ctDNA was detectable in 8/30 cases (26%). In the prospective cohort, ctDNA was detectable in 17/25 (68%) patients that developed metastases. In the metastatic cohort, ctDNA was detectable in 6/6 (100%).	ctDNA levels in primary UM are not associated with survival, but this was the strongest predictor of OS in MetUM. Decreases in ctDNA levels are an indicator of response to immunotherapy.
de Bruyn et al. [14]	PB	Primary and metastatic	*n* = 34; for ctDNA detection (*n* = 18) and/or SCNA analysis (*n* = 26) at various time points.	ddPCR and sWGS	ctDNA was detectable in 38% (5/13) of patients at diagnosis, in 77% (10/13) upon detection of metastatic disease, and in 50% (3/6) during fSRT. Loss of Chr 3 was detected in 70% (7/10) of patients with MetUM.	No SCNA profiles and ctDNA levels were low or undetectable in localized disease. ctDNA levels in metastatic patients could be a biomarker of disease progression.
Nell et al. [20]	Vitreous fluid	Primary	*n* = 65.	ddPCR	39/65 (60%) patients had Gαq signaling mutations; 13/15 (87%) had a BSE mutation; Chr 3p losses were detected in 13/15 (87%) samples; Chr 8q gains were identified in 15/17 samples (88%).	cfDNA was associated with larger tumors of BSE mutation. and CNA results could be inferred from vitreous fluid liquid biopsy.
Park et al. [85]	PB	Metastatic	*n* = 17; samples were collected at baseline, EDT and on-treatment.	ddPCR and NGS	At baseline, ctDNA was detected by ddPCR in 94% (16/17) of patients, by NGS in 88% (15/17), and for those on treatment it was identfied by NGS in 94% (16/17).	Absolute level of EDT ctDNA is indicative of treatment response. Increasing UM ctDNA preceded radiological progression.
Carvajal et al. [92]	PB	Metastatic	*n* = 127	Multiplex PCR and NGS	80% (94/118) had mutations detected in one or more genes (Gαq and *SF3B1*).	ctDNA as an early indicator of clinical benefit from tebentafusp; Post-treatment reduction in ctDNA levels correlated with survival benefit.
Kim et al. [13]	PB	Primary	*n* = 26; samples were collected at various time points.	NGS	In 31% (8/26) of patients, ctDNA was detected during or after brachytherapy. No ctDNA was detected in any of the samples collected before brachytherapy.	Brachytherapy increases the presence of ctDNA. The allele fraction detected correlates with the largest basal diameter and tumor thickness.
Ny et al. [93]	PB	Metastatic	*n* = 29; samples were collected at various time points.	NGS	75% (12/16) of patients with PD and 37% (3/8) with stable disease had detectable ctDNA levels.	Low baseline ctDNA levels predicted long OS but not PFS. The ctDNA levels were lower (not significantly) in patients with PR vs. those with PD.
Sato et al. [98]	PB	Primary and metastatic	14 MetUM and two non-metastatic patients; samples collected at various time points	ULP-WGS	78% (11/14) of patients with MetUM had detectable ctDNA. 8q gain was detected in all; Loss of Chr 3 was detectable in 59% (10/17).	ctDNA in metastatic patients can be detected by ULP-WGS, and ctDNA levels correlate with disease status.

AH—aqueous humor; CB—ciliary body; PB—peripheral blood; fSRT—fractionated stereotactic radiotherapy treatment, ddPCR—digital droplet PCR; NGS—next-generation sequencing; ULP-WGS—ultra-low pass whole-genome sequencing; sWGS—shallow whole-genome sequencing; EDT—early during treatment; OS—overall survival; PFS—progression-free survival; PR—partial response; PD—progressive disease; MetUM—metastatic UM; Chr—chromosome.

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
