# Peer review of "Optimizing ctDNA: An Updated Review of a Promising Clinical Tool for the Management of Uveal Melanoma"

_cancers, 2024, doi:10.3390/cancers16173053_

Round 1

Reviewer 1 Report

Comments and Suggestions for Authors

This is an excellent review from Varela et al. about the potential use of ctDNA as clinical tool in uveal melanoma.

Minor comments:

- In the manuscript text, please cite “(as reviewed in [xx])” for references corresponding to reviews.

- Graphical Abstract: In the eye image on the left, it would be more pertinent to localize part of the tumor in the choroid layer (it seems to float in the vitreous).

- Reference #19: Is this allowed to cite a manuscript that has yet to be evaluated (deposited in medRxiv)?

- Lines 170-171: Active secretion of ctDNA is also possible from viable cells via extracellular vesicles or lipoprotein complexes.

- Table 1: If the sensitivity and specificity values for the “ddPCR + LNA-Clamp ddPCR” and “IC3D ddPCR” techniques were not determined, please indicate NA (not available) in these columns.

- Please comment in section 5 if these molecular techniques for ctDNA detection were previously used to analyze uveal melanoma ctDNA.

- Verify if this reference could be added to the review: Anal Chem. 2023 Nov 14;95(45):16692-16700 (PMID: 37921444).

Comments on the Quality of English Language

- Lines 53-54: Correct for “circulating tumor cells (CTCs)”.

- Lines 55, 214: Use the term “extracellular vesicles” rather than “exosomes”.

- Line 76: Please define IVDR (In Vitro Diagnostic Regulation?).

- Lines 80-82: I think it will be more accurate to replace “cells” by “analytes” at the end of this sentence (“ctDNA-based tests are more common than CTC-based tests because more technologies are available to isolate these cells”) since ctDNA is not a cell.

- Line 109: Correct for “A recent study by de Bruyn et al. showed”.

- Lines 309-310: Correct for “Thus, the presence or absence of the PCR product determines the diagnosis”.

- Lines 360-361: Correct for “single nucleotide variants (SNVs)”.

- Line 438: Correct for “The detection of SCNAs and gene mutations”.

- Line 442: Correct for “de Bruyn et al. [13] showed”.

- Lines 454-455: Italicize the BAP1 and GNAQ gene symbols.

- Table 3: In column titles, replace “Sample type » by « Fluid type ». Correct for “SCNA analysis” at line de Bruyn et al./column number of patients. Replace “pts” by “patients” at line Beasley et al./column detection rate.

- Line 476: Correct for “Liquid biopsy is increasingly used to for the molecular characterization”.

- Line 490: Italicize the KRAS gene symbol.

- Line 502: Correct “into the bloodstream”.

Author Response

Thank you very much for taking the time to review this manuscript. Please find the detailed responses below (in blue) and the corresponding corrections implemented in the re-submitted files.

Regarding minor comments:

  1. In the manuscript text, please cite “(as reviewed in [xx])” for references corresponding to reviews. In reviews used as reference we cited as you advise us.
  2. Graphical Abstract: In the eye image on the left, it would be more pertinent to localize part of the tumor in the choroid layer (it seems to float in the vitreous). Thank you very much for your comment and we have taken it into account, but unfortunately it was not possible to change the exact position of the image composition.
  3. Reference #19: Is this allowed to cite a manuscript that has yet to be evaluated (deposited in medRxiv)? We think so. According to NIH: "The NIH encourages investigators to use interim research products, such as preprints, to speed the dissemination and enhance the rigor of their work.” In addition, there are instructions how to cite a preprint: "Preprints deposited in medRxiv should be cited using their digital object identifier (DOI)."
  4. Lines 170-171: Active secretion of ctDNA is also possible from viable cells via extracellular vesicles or lipoprotein complexes. Thank you very much for your comment, but in this particular case we would like to focus on ctDNA release after radiotherapy treatment.
  5. Table 1: If the sensitivity and specificity values for the “ddPCR + LNA-Clamp ddPCR” and “IC3D ddPCR” techniques were not determined, please indicate NA (not available) in these columns. DONE
  6. Please comment in section 5 if these molecular techniques for ctDNA detection were previously used to analyze uveal melanoma ctDNA. We have mentioned in the most common techniques used to detect alterations in  uveal melanoma (UM). If the technique has not been used in UM or has rarely been used we prefer not to comment. We have introduced some mentions of the most common techniques used in UM. If the technique has not been used in UM or has been rarely, we have considered that it would be better not to comment on it at all.
  7. Verify if this reference could be added to the review: Anal Chem. 2023 Nov 14;95(45):16692-16700 (PMID: 37921444). ADDED

Regarding Comments on the Quality of English Language all the corrections have been implemented.

- Lines 53-54: Correct for “circulating tumor cells (CTCs)”. 

- Lines 55, 214: Use the term “extracellular vesicles” rather than “exosomes”.

- Line 76: Please define IVDR (In Vitro Diagnostic Regulation?).

- Lines 80-82: I think it will be more accurate to replace “cells” by “analytes” at the end of this sentence (“ctDNA-based tests are more common than CTC-based tests because more technologies are available to isolate these cells”) since ctDNA is not a cell.

- Line 109: Correct for “A recent study by de Bruyn et al. showed”.

- Lines 309-310: Correct for “Thus, the presence or absence of the PCR product determines the diagnosis”.

- Lines 360-361: Correct for “single nucleotide variants (SNVs)”.

- Line 438: Correct for “The detection of SCNAs and gene mutations”.

- Line 442: Correct for “de Bruyn et al. [13] showed”.

- Lines 454-455: Italicize the BAP1 and GNAQ gene symbols.

- Table 3: In column titles, replace “Sample type » by « Fluid type ». Correct for “SCNA analysis” at line de Bruyn et al./column number of patients. Replace “pts” by “patients” at line Beasley et al./column detection rate.

- Line 476: Correct for “Liquid biopsy is increasingly used to for the molecular characterization”.

- Line 490: Italicize the KRAS gene symbol.

- Line 502: Correct “into the bloodstream”.

Reviewer 2 Report

Comments and Suggestions for Authors

I would like to congratulate Varela et al for the manuscript "Optimizing ctDNA: an Updated Review of a Promising Clinical Tool for the Management of Uveal Melanoma" which represents an excellent summary of the potential use and current challenges on liquid biopsies for uveal melanoma. The manuscript is well written and documented. 

Line 21 and line 25: "Uveal melanoma (UM) is the most common primary intraocular tumor in adults". Needs correction. Uveal melanoma is the most common primary malignant intraocular tumor in adults

Line 25: "Tissue sampling also poses a risk of extraocular dissemination" needs reference for this statement

Comments on tumors' size and more accurate results in aqueous samples and vitreous samples

Author Response

Thank you very much for taking the time to review this manuscript. Please find the detailed responses below (in blue) and the corresponding corrections implemented in the re-submitted files.

Regarding Comments and Suggestions for Authors:

1. Line 21 and line 25: "Uveal melanoma (UM) is the most common primary intraocular tumor in adults". Needs correction. Uveal melanoma is the most common primary malignant intraocular tumor in adults. We want to thank you for the comment. This modification has been implemented in the manuscript.

2. Line 25: "Tissue sampling also poses a risk of extraocular dissemination" needs reference for this statement. We have modified the position of this sentence from the Abstract to the Introduction on line 66. in addition we have included a reference.

3. Comments on tumors' size and more accurate results in aqueous samples and vitreous samples. We really appreciate your comment, but in this article we would like to focus on molecular techniques and try to provide an overview of their use in clinical routine for all fluids.

Reviewer 3 Report

Comments and Suggestions for Authors

This is a clearly written and comprehensive review of the use of liquid biopsy in the diagnosis and management of uveal melanoma with a good description and critique of available methodologies. it is a useful review for Ophthalmologists and Ocular  Oncologists.

Author Response

Thank you very much for taking the time to review this manuscript. We really appreciatte very much your comments.